# NAD^+^ Precursors Reverse Experimental Diabetic Neuropathy in Mice

**DOI:** 10.3390/ijms25021102

**Published:** 2024-01-16

**Authors:** Krish Chandrasekaran, Neda Najimi, Avinash R. Sagi, Sushuma Yarlagadda, Mohammad Salimian, Muhammed Ikbal Arvas, Ahmad F. Hedayat, Yanni Kevas, Anand Kadakia, Tibor Kristian, James W. Russell

**Affiliations:** 1Department of Neurology, University of Maryland School of Medicine, Baltimore, MD 21201, USA; krishchandra0630@gmail.com (K.C.); neda_najimi@yahoo.com (N.N.); ysushuma@gmail.com (S.Y.); m.ikbalarvas@gmail.com (M.I.A.); fahim.hedayat@yahoo.com (A.F.H.); ykevas@terpmail.umd.edu (Y.K.); akadakia1991@gmail.com (A.K.); 2Department of Anesthesiology, University of Maryland School of Medicine, Baltimore, MD 21201, USA; tkristian@som.umaryland.edu; 3Veterans Affairs Medical Center, Baltimore, MD 21201, USA; 4CAMC Institute for Academic Medicine, 415 Morris Street Suite 300, Charleston, WV 25301, USA

**Keywords:** streptozotocin, high fat diet, diabetic neuropathy, NAD^+^, mitochondria, SIRT1, NEDD4-1

## Abstract

Abnormal NAD^+^ signaling has been implicated in axonal degeneration in diabetic peripheral neuropathy (DPN). We hypothesized that supplementing NAD^+^ precursors could alleviate DPN symptoms through increasing the NAD^+^ levels and activating the sirtuin-1 (SIRT1) protein. To test this, we exposed cultured Dorsal Root Ganglion neurons (DRGs) to Nicotinamide Riboside (NR) or Nicotinamide Mononucleotide (NMN), which increased the levels of NAD^+^, the SIRT1 protein, and the deacetylation activity that is associated with increased neurite growth. A SIRT1 inhibitor blocked the neurite growth induced via NR or NMN. We then induced neuropathy in C57BL6 mice with streptozotocin (STZ) or a high fat diet (HFD) and administered NR or NMN for two months. Both the STZ and HFD mice developed neuropathy, which was reversed through the NR or NMN administration: sensory function improved, nerve conduction velocities normalized, and intraepidermal nerve fibers were restored. The NAD^+^ levels and SIRT1 activity were reduced in the DRGs from diabetic mice but were preserved with the NR or NMN treatment. We also tested the effect of NR or NMN administration in mice that overexpress the SIRT1 protein in neurons (nSIRT1 OE) and found no additional benefit from the addition of the drug. These findings suggest that supplementing with NAD+ precursors or activating SIRT1 may be a promising treatment for DPN.

## 1. Introduction

The regulation of Nicotinamide Adenine Dinucleotide (NAD^+^) has been shown to be critical in the prevention and reversal of diabetic polyneuropathy (DPN) and diabetic cognitive impairment [1,2,3,4,5]. NAD^+^ levels decline with age in individuals with neurodegenerative conditions, acute brain injury, obesity, or diabetes [6,7]. The loss of NAD^+^ results in the impairment of mitochondrial and cellular function and can result in axonal degeneration [8,9]. NAD^+^ can induce a response in the peripheral nervous system through several pathways including sirtuins, poly (ADP-ribosyl) transferase 1 (PARP1), and cluster of differentiation 38 (CD38) [7,10,11]. The activation of the NAD^+^ hydrolase enzyme sterile alpha and TIR motif constraining 1 (SARM1) or the deleted in bladder cancer protein 1 (DBC1) can inhibit SIRT1 activity, while the activation of PARP1 or CD 38 promotes axonal degeneration [12,13,14]. In contrast, knockout of PARP1, CD38, SARM1, and DBC1 protects mice against neuropathy induced via a high fat diet (HFD) or chemotherapy-induced neuropathy [14,15,16,17,18]. The proteins that resynthesize NAD^+^ via the salvage pathway, such as nicotinamide mononucleotide adenylyltransferase 1–3 (NMNAT1–3), protect against axonal degeneration [19,20].

Among the NAD^+^ consuming enzymes, sirtuin–1 (SIRT1) is unique because the activation of SIRT1 protects against diabetic polyneuropathy (DPN). In contrast, the inhibition of other NAD^+^ consuming proteins protects against DPN [3,21]. Although it is not clear which one of these pathways is a major contributor to peripheral neuropathy induced via Streptozotocin (STZ) or an HFD, what seems to be common is that an increase in cellular NAD^+^ levels is likely to be protective [2].

Sirtuins cleave NAD^+^ into nicotinamide and 1′-O-acetyl-ADP-ribose [22] or 2′- and 3′-O-acetyl-ADP-ribose [23] with a corresponding deacetylation of lysine residues. The SIRT1 activity requires NAD^+^ [24]. This causes a decrease in PGC-1α acetylation and an increase in PGC-1α activity [3,5]. Overall, there is evidence that SIRT1 is neuroprotective and that knockdown of SIRT1 increases susceptibility to neurodegeneration [2,3,5,25,26,27,28,29,30]. In part, SIRT1 neuroprotection may be because of improved mitochondrial function and energy homeostasis [2,3,31]. The SIRT1 activity in the nucleus deacetylates transcriptional and co-transcriptional factors that regulate glucose homeostasis and fat oxidation. Furthermore, there is evidence that the activation of SIRT1 is important in axonal regeneration [21].

Interventions with key NAD^+^ intermediates, such as nicotinamide mononucleotide (NMN) and nicotinamide riboside (NR), have drawn significant attention in the fields of aging, diabetes, obesity, and metabolic syndrome because these supplements can increase and replenish cellular NAD^+^ levels. NMN is synthesized from nicotinamide, a form of water-soluble vitamin B3, and phosphoribosyl-1-pyrophosphate (PRPP) by NAMPT, the rate-limiting NAD^+^ biosynthetic enzyme in mammals. NMN is also synthesized from NR via a nicotinamide riboside kinase- (NRK-) mediated phosphorylation reaction. The conversion of NMN into NAD^+^ is catalyzed by nicotinamide mononucleotide adenyl transferases (NMNATs). The conversion of NR by NRK suggests that NR could be administered orally to increase NAD^+^ levels. On the other hand, the transport of NMN is unclear. Therefore, we used intraperitoneal administration of NMN. More recently a potential transporter of NMN has been identified [32]. Therefore, NMN is also orally bioavailable. The systemic administration of NMN or NR effectively increases NAD^+^ levels in various peripheral tissues [1].

We hypothesized that the administration of NAD^+^ precursors nicotinamide mononucleotide (NMN) or nicotinamide riboside (NR) would reverse peripheral neuropathy induced via STZ or an HFD. We explored the potential mechanisms of the action of the NAD^+^ precursors on neurite growth and on SIRT1 expression in DRG neurons. Our results show that NMN/NR administration increased the NAD^+^ levels in DRG neurons, increased neurite growth, improved nerve function, and reversed the loss of IENFD. The results suggest that supplementation with NAD+ precursors may benefit patients with DPN.

## 2. Results

### 2.1. NR or NMN Increases Neurite Length through a SIRT1-Mediated Mechanism in Immortalized DRG Neuronal Cultures

We generated a doxycycline-inducible transgenic mouse model overexpressing SIRT1 protein in neurons (nSIRT1OE), including dorsal root ganglion neurons (DRGs) [3]. We cultured adult DRG neurons from 3-month-old nSIRT1OE mice and wild-type (WT) mice for 4 days and measured the length of neurite outgrowth. Neurite outgrowth was defined as the total amount of neurite extension in μm per neuron, and the average value was calculated for each culture. There was a twofold increase in the neurite length in cultured primary DRG neurons from the nSIRT1OE mice compared to the WT mice (from 200 ± 25 μm in WT to 410 ± 45 μm in nSIRT1OE), which corresponded to elevated SIRT1 protein levels and SIRT1 deacetylase activity (Figure 1A).

Previous studies have reported that the administration of NAD^+^ precursors can enhance sirtuin pathways [21,33,34]. The administration of NR can elevate NAD^+^ levels and increase the activity of nuclear and mitochondrial NAD^+^-dependent protein lysine deacetylases, including SIRT1 and SIRT3. Similarly, NMN treatment can ameliorate insulin resistance induced via a high fat diet by restoring NAD^+^ biosynthesis and SIRT1/SIRT3 activity [33,35]. We investigated the relationship between NAD^+^ and SIRT1 expression and neurite outgrowth using immortalized rat DRG neurons (50B11). We differentiated the cultures with forskolin to induce neurite outgrowth in the presence or absence of added NAD^+^ precursors. We also added SIRT1 inhibitor EX527 (5 μM) to duplicate cultures to study the role of SIRT1 in neurite outgrowth. Neurite length was measured after 72 h of treatment. The addition of NMN or NR increased neurite length compared to the vehicle control (*p* < 0.01). However, the addition of EX527 inhibited forskolin-induced neurite growth in NR- or NMN-treated cultures (*p* < 0.001) (Figure 1B). Furthermore, the overexpression of SIRT1 using an overexpression (OE) vector increased neurite growth, which was dependent on SIRT1 activity and blocked by EX527 (Figure 1B). 

### 2.2. Dietary Administration of NR or Subcutaneous Administration of NMN and Peripheral Neuropathy Induced via Streptozotocin (STZ) in Mice

In this study, we aimed to investigate whether NR would reverse DPN in STZ diabetic mice. To test this hypothesis, we used 2-month-old, male, C57Bl6 STZ-induced diabetic mice (*n* = 20) and wild-type (WT), non-diabetic, C57Bl6 mice (n = 10). Baseline neuropathy measurements were taken in both groups (Figure 2, Time point 0), and then all mice were fed a control diet (Harlan Teklad) for 2 months. 

At the end of 2 months, neuropathy measurements were made in all mice. Once neuropathy developed, based on measurements of the neuropathy, the diabetic mice were split into two groups: diabetic mice that received NR in their diet (300 mg/kg) for an additional 2 months (Group #3) and diabetic mice that continued their normal diet (Group #2) for an additional 2 months. Meanwhile, the non-diabetic mice continued as the control diet (Group #1) for a total period of 4 months. At 4 months, nerve conduction studies and mechanical allodynia (MA) were measured in all three groups of mice.

The results showed that at 2 months the STZ-induced diabetic mice compared to the non-diabetic mice had significantly slower sensory motor nerve conduction velocity (SMNCV) (from 43 ± 6 m/s in the non-diabetic mice to 27 ± 7 m/s in the STZ mice; *p* < 0.001), decreased thermal mechanical latency (TML) (from 2.2 ± 0.3 msec to 1.2 ± 0.2 msec; *p* < 0.001), decreased tail sensory nerve conduction velocity (TSNCV) (from 41 ± 3 m/s to 27 ± 5 m/s; *p* < 0.001), and had developed mechanical allodynia (MA) (Figure 2). However, the dietary supplementation of NR to the STZ-induced diabetic mice for 2 months resulted in normal tactile allodynia at 4 months, while the von Frey paw withdrawal threshold was still decreased in the STZ mice (Figure 2). Moreover, hind paw skin biopsies at 4 months showed that the intraepidermal nerve fiber density (IENFD) was significantly decreased in the STZ mice compared to the non-diabetic mice. In contrast, the IENFD was higher in the STZ + NR mice compared to the STZ mice (STZ = 12 ± 2 fibers/mm vs. STZ + NR = 24 ± 3 fibers/mm; *p* < 0.001). These findings suggest that the dietary administration of NR could effectively reverse peripheral neuropathy in STZ-induced diabetes. 

We then investigated whether subcutaneous administration of NMN could reverse DPN in STZ diabetic mice. The schematic for the experiment and data at 4 months is shown in Table 1. Subcutaneous (sc) administration of NMN on alternate days at a dose of 100 mg/kg reversed the STZ-induced changes in the nerve conduction parameters. The administration of NMN to STZ mice reversed tactile allodynia at 4 months. Hind paw skin biopsies showed a significant decrease in the IENFD in the STZ mice compared to the non-diabetic mice. In contrast, the IENFD at 4 months was higher in the STZ + NMN mice compared to the STZ mice (STZ = 12 ± 2 fibers/mm vs. STZ + NMN 100 = 22 ± 3; *p* < 0.001). There was no significant change in the IENFD in the non-diabetic mice compared to the non-diabetic + NMN mice. These results suggest that subcutaneous administration of NMN reverses STZ-induced peripheral neuropathy.

### 2.3. Dietary Administration of NR or Subcutaneous Administration of NMN Reverses HFD-Induced Peripheral Neuropathy in Mice

We tested whether the dietary administration of NR could reverse HFD-induced peripheral neuropathy. After 2 months of HFD feeding (n = 8), the C57BL6 mice showed significant abnormalities in the NCSs, which were consistent with the development of peripheral neuropathy (Figure 3A–C). After 2 months on an HFD, there was a significant decrease in the Von Frey paw withdrawal threshold in the HFD mice compared to the CD mice that persisted up to 16 weeks and is consistent with developing tactile allodynia (Figure 3D). In contrast, the HFD + NR mice had preserved NCVs and Von Frey thresholds, consistent with protection against peripheral neuropathy through dietary administration of NR (Figure 3). Two months after feeding mice with either a CD or an HFD, the mice were euthanized and the paw skins were examined for the IENFD. Skin biopsies showed a significant decrease in the IENFD of the HFD mice (14.5 ± 2.5 fibers/mm) compared to the CD mice (29.6 ± 4.4 fibers/mm; *p* < 0.001). In contrast, in mice that were then fed NR for a further 2 months, the IENFD was the same as the CD mice (HFD + NR 150 mg/kg = 27.6 ± 3.5 fibers/mm; HFD + NR 300 mg/kg = 30.8 ± 3.2 fibers/mm) (Table 2). These results suggest that the dietary administration of NR protected against HFD-induced peripheral neuropathy.

In HFD C57BL6 mice, the SMNCV decreased from 42.77 ± 2.09 m/s in the CD to 29.88 ± 1.23 m/s in the HFD; *p* < 0.001. The TML increased from 1.3 ± 0.07 ms to 2.89 ± 0.3 ms; *p* < 0.001. The TSNCV decreased from 35.1 ± 0.9 m/s in the CD to 29.34 ± 1.34 m/s in the HFD; *p* < 0.001. The changes in the NCVs in the HFD mice were consistent with the development of peripheral neuropathy. After 8 weeks of an HFD, there was a significant decrease in the von Frey paw withdrawal threshold in the HFD mice compared to the CD mice (CD = 1.35 ± 0.24 g vs. HFD = 0.68 ± 0.12 g; *p* < 0.001), which remained the same up to 16 weeks, consistent with the development of tactile allodynia. Skin biopsies showed a significant decrease in the IENFD of the HFD mice (14.5 ± 2.5 fibers/mm) compared to the CD mice (29.6 ± 4.4 fibers/mm; *p* < 0.001). After confirming that the HFD mice had developed neuropathy as observed by the changes in the NCSs and MA, the HFD mice were injected with NMN (100 mg/kg) intraperitoneal (ip) on alternate days for an additional 2 months. The schematic approach is shown in Table 2A. The results of the NCSs are shown in Table 2B. Statistical comparisons were made among the three groups. The administration of NMN reversed all the deficits of HFD-induced neuropathy. The administration of NMN to non-diabetic mice had no significant effect [1].

### 2.4. Administration of NMN or NR Corrects Alterations Induced via STZ or HFD in the NAD^+^ Metabolome

We hypothesized that T1D and T2D might alter the NAD^+^ metabolome in the DRGs and contribute to neuropathy. We therefore employed LC-MS/MS to measure the NAD^+^ metabolome in the DRGs from freshly euthanized mice (Figure 4). We examined how NR administration influenced the NAD^+^ biosynthesis in the brains and DRGs of HFD mice by measuring the levels of NAD^+^ and its metabolites. After 2 months of feeding the HFD mice with NR (300 mg/kg/day), the levels of NAD^+^ were increased (HFD = 1390 ± 81 vs. HFD + NR = 2140 ± 79 pmol/mg protein; *p* < 0.01). A smaller but not significant increase in the precursors (NR, NMN) and the degradation product nicotinamide (NAM) was observed in the NR-treated and the HFD mice [1,2].

### 2.5. No Additive Effect of NMN Administration in nSIRT1OE Mice in the Reversal of HFD-Induced Neuropathy in C57BL6 Mice

The nSIRT1OE transgenic mice were generated by crossing a CamK2a-tTA mouse with a TRE-SIRT1/mito-eYFP mouse, creating a Tet-Off mouse [3]. The expression of nSIRT1 is induced in the absence of doxycycline in the diet and is shut off with doxycycline in the diet (DOX, 200 mg/kg diet). SIRT1 uses NAD^+^ to deacetylate its substrates. Therefore, we tested whether there is an additive effect of nSIRT1OE with added NMN. Three-month-old, DOX-fed, male, nSIRT1OE-OFF mice were fed with either a CD or an HFD for 2 months. NCSs were completed at the beginning and at the end of 2 months. The results are shown in Figure 5 and Table 3. Feeding mice with an HFD-induced neuropathy as determined by the NCSs. After 2 months of HFD feeding, nSIRT1OE was turned on by removing the DOX from the diet. The nSIRT1OE mice were divided into two groups: one group (Group #3) was fed with a DOX-free HFD for an additional 2 months; another group (Group #4) was fed with a DOX-free HFD, and NMN was administered on alternate days at a dose of 100 mg/kg ip for an additional 2 months. NCSs were performed after 2 months of the administration of NMN. nSIRT1OE alone or with NMN treatment reversed all the deficits of the HFD-induced neuropathy.

## 3. Discussion

### 3.1. Effect of NAD Precursors on Neurite Growth

This present study shows that NMN and NR can directly induce neurite outgrowth in neuronal culture, and this is blocked by the SIRT1 inhibitor EX527. Nicotinamide phosphoribosyl transferase (NAMPT) is the rate-limiting enzyme for NAD^+^ salvage synthesis, generating NMN from nicotinamide, and it may serve as a therapeutic target to restore adult neurogenesis [36]. NAD^+^ is synthesized from NR following its phosphorylation to NMN by the ATP-dependent enzyme, nicotinamide riboside kinase. NMN is converted to NAD^+^ via NMNAT. Axons require the axonal NAD-synthesizing enzyme NMNAT2 to survive [20,37]. NMNAT2 is reduced in the brains of Alzheimer’s and other neurodegenerative diseases [38]. In cortical neurons, knockdown of NMNAT1 results in a significant decrease in axon growth and branching [39]. However, NMNAT2 alone is expressed in peripheral axons and appears to be critical in maintaining axonal integrity. The depletion of NMNAT2 in vitro may result in the accumulation of NMN that failed to be converted to NAD^+^, and this is associated with more rapid axonal degeneration post-transection in vitro [37,40]. The depletion of NMNAT2 can trigger axonal degeneration or may result in defective axon growth potentially because of the accumulation of the NMN. Interestingly, NMN deamidase, a bacterial enzyme, which, like NMNAT2, consumes NMN but, unlike NMNAT2, does not synthesize NAD, delays axon degeneration in primary neuronal cultures [20]. Furthermore, a low-level expression of NMN deamidase may reduce the accumulation of NMN in injured mouse sciatic nerves, preserve some axons for up to three weeks, and rescue axonal growth in mice that lack NMNAT2. In this present study, increased levels of NMNAT2 RNA are directly associated with a decrease in the IENFD from human skin biopsies from patients with diabetic neuropathy, further supporting the role of NMNAT2 in axonal regeneration.

High levels of NAD^+^ can activate sirtuins [41], specifically SIRT1 [3], and prevent a decrease in the IENFD in experimental diabetic neuropathy [1]. In this present study, the overexpression of SIRT1 is directly able to induce neurite outgrowth, and this effect is blocked by the inhibition of SIRT1. SIRT can specifically deacetylate as well as colocalize with the Ubiquitin E3 protein ligase NEDD4-1 [3], which is required for axonal growth and regeneration [42,43,44,45,46,47].

### 3.2. Regulation of Axonal Growth and Mitochondrial Function via SIRT1-Mediated Deacetylation of NEDD4-1

The SIRT1 activity in the nucleus deacetylates transcriptional and co-transcriptional factors that regulate glucose homeostasis and fat oxidation. However, there is a distinct mitochondrial isoform [48,49] that may also play a role in mitochondrial regulation. However, SIRT1 also has a role in axonal growth. The E3-Ubiquitin ligases control the penultimate step in the ubiquitin pathway and control both the efficiency and substrate specificity of the ubiquitination reaction.

Although other E3-ligases have been implicated in sensory axonal degeneration, only neural precursor cell expressed developmentally down-regulated protein 4 (NEDD4-1) is deacetylated by SIRT1 [3]. NEDD4-1 is involved in various basic cellular functions [50]. Most importantly, NEDD4-1 is a crucial modulator of axonal and dendritic growth, which is required for the proper formation and function of neuromuscular junctions [45,51,52,53]. The disruption of the E3 ligase NEDD4-1 in the DRG neurons interrupts axon outgrowth [54]. NEDD4-1 deacetylation also has an important role in preventing neurodegenerative processes by reducing misfolded protein aggregates and synuclein aggregation [55]. Earlier studies showed that PTEN could be one of the substrates of NEDD4-1, but recent results suggest that NEDD4-1, the serine/threonine kinase TNIK, and Rap2A form a complex that controls the NEDD4-1-mediated ubiquitination of Rap2A [44]. In NEDD4-1 knockdown cells, abnormal mitochondria have been observed, suggesting that the activation of NEDD4-1 promotes mitophagy and autophagy via ubiquitination of the protein sequestosome 1, which leads to the degradation of abnormal mitochondria [42,52,56,57]. This is consistent with our observation that NR or NMN increases the levels of NAD^+^ and the SIRT1 protein and promotes neurite growth.

### 3.3. NMNAT2, SARM, and Axonal Degeneration

Axonal degeneration and regeneration are controlled through sterile alpha and Toll/interleukin-1 receptor motif-containing 1 (SARM1) through a well-regulated system comprising NAD^+^ and NMN. SARM1, when activated, induces axonal degeneration [12]. The NAD^+^ binds to an allosteric site within the ARM domain of SARM1 and locks the ARM and TIR domains to an inactive state. Exogenous NAD^+^ precursors (NR or NMN) utilize the axonal enzyme NMNAT2 to increase the NAD^+^ levels and to promote axonal maintenance and growth. In contrast, NMN binds to the same NAD^+^ site of SARM1 and disrupts the ARM-TIR lock, freeing the TIR domains to associate into an activated NAD^+^ hydrolyzing (NADase) enzyme, thus activating axonal degeneration. The synthesis of axonal NAD^+^ is regulated by the enzyme NMNAT2. The NMNAT2 converts NMN to NAD^+^, using one ATP. However, the half-life of NMNAT2 in axons appears to be short, and it requires transport from the cell body. The disruption of axonal transport leads to the depletion of NMNAT2 in distal axons, thus promoting degeneration. This leads to the accumulation of NMN, which is further aggravated by the salvage pathway in which the NAD^+^ is converted to nicotinamide and in turn to NMN by the NAMPT. This explains the apparent paradox in which NMN can result in regeneration of intact but not injured axons in vivo: NMNAT2 can be transported to distal axons, NMN can be converted to NAD^+^, and SARM1 is inhibited. In contrast, in transected axons in vitro, the transport of NMNAT2 is inhibited, NMN accumulates, SARM1 is activated, and axonal degeneration occurs. Importantly, a SARM1 deficiency provides lifelong rescue against NMNAT2 deletion [58,59] and also prevents demyelination post-axonal injury [60].

Interestingly, SARM1 also has a mitochondrial targeting sequence, can disrupt mitochondrial dynamics, and can bind to and stabilize PTEN-induced putative kinase 1 (PINK1), thus inducing mitophagy [61]. However, deletion of the mitochondrial localization sequence does not alter the ability of SARMs to promote axon degeneration after axotomy. Thus, the mitochondrial targeting sequence may provide a mechanism to eliminate cellular SARM1 through mitophagy. The mitochondrial localization of SARM1 complements the coordinated activity of NMNAT2 that promotes axonal survival. Thus, a SARM1 inhibitor coupled with either NR or NMN may be more effective than a single agent alone in preventing or treating diabetic neuropathy.

### 3.4. NAD Precursors and Axonal Transport

Distal axon survival is dependent on axonal transport that further requires energy. Thus, in diabetic neuropathy where there is impaired metabolic function, there is an increased risk for axonal degeneration and neuronal death. We have previously shown that the NAD^+^ dependent deacetylase enzyme Sirtuin 1 (SIRT1) can prevent the activation of these pathways and promote axonal regeneration [3].

NMNAT2 is critical for axon survival in primary culture, and its depletion may contribute to axon degeneration. In the absence of NMNAT2, there is the failure of axonal growth. NMNAT2 is found in mouse axonal vesicles that undergo fast axonal transport [62]. If the NMNAT2 loses association with the vesicles, the protein half-life is increased and boosts the axon protective capacity. A similar mechanism exists in vivo in mouse sciatic nerves and Drosophila olfactory axons. Furthermore, reducing the activity of the SARM1 can reduce axonal transport deficits and suppress axon degeneration in NMNAT2 KO neurons [63]. Microtubules are dynamic cytoskeletal polymers that are important in axonal transport. NAD and NMNAT2 appear to influence microtubule growth through indirect mechanisms [64]. Overall, NAD and NMNAT2 ensure axonal health, ensure efficient vesicular glycolysis, and promote fast axonal transport.

### 3.5. NAD Precursors Affect Lipid Regulation but Have No Significant Effect on Glucose Regulation

In this present study, both NR and NMN lower the levels of lipids and specifically triglycerides. The metabolic factors resulting in diabetic neuropathy are complex but include both hyperglycemia and hyperlipidemia [65]. NA/NAM has been used to treat dyslipidemia through lowering the levels of triglycerides and LDL and raising the level of HDL [66]. In the current study, inducing diabetes with STZ or feeding with an HFD caused a significant increase in plasma glucose, cholesterol, triglycerides, and NEFA. NR and NMN had no significant effect on glucose levels in HFD or STZ mice but reduced levels of cholesterol and triglycerides. Furthermore, NR treatment did not affect the GTT similar to obese men where NR has no effect on glycemic parameters [67]. The decrease in lipid levels suggests that both NMN and NR may promote lipid oxidation. NR or NMN treatment increases the SIRT1 protein levels and upregulate genes that are involved in the regulation of fatty acid metabolism [68,69]. Similar results have been shown in humans using a meta-analysis based on published clinical trials in humans that showed supplementation with NAD^+^ precursors, compared with a placebo or no treatment, improved triglycerides, total cholesterol, LDL, and HDL levels but did not alter hyperglycemia [70]. NAD^+^ precursors have been shown to exert their effect on lipids through sirtuin regulation of mitochondrial beta-oxidation [71]. Experimental nonalcoholic fatty liver disease (NAFLD) lowers hepatic NAD^+^ levels, reducing hepatic mitochondrial content, function, and ATP levels. In contrast, NR added to the diet reverses NAFLD through inducing a SIRT1 and SIRT3—a dependent mitochondrial unfolded protein response that increases hepatic beta-oxidation and mitochondrial complex content and activity.

### 3.6. There Is No Additive Effect on Neuropathy with SIRT1 Overexpression

This present study showed that NR and NMN had no additive neuroprotective effect to SIRT1 alone. The sirtuins use NAD^+^ as a co-substrate to catalyze the deacetylation and/or mono-ADP ribosylation of target proteins. Overall, there is evidence that SIRT1 is neuroprotective and that knockdown of SIRT1 increases susceptibility to neurodegeneration [2,3,5,25,26,27]. In part, SIRT1 neuroprotection may be because of improved mitochondrial function and energy homeostasis [2,3,31]. High levels of NAD^+^ can activate sirtuins and, in turn, prevent oxidative injury in neurons [1,5,26]. Once NAD^+^ is used by the sirtuins, nicotinamide (Nam) is generated as a product. The salvage pathway converts Nam back to NMN through the enzyme nicotinamide phosphoribosyl transferase (NamPRT). Neurons are particularly susceptible to depletion of NAD owing to their apparent lack of a fully functional salvage pathway. Thus, a combination of NR or NMN with SIRT1OE would likely only be beneficial if there was neuronal depletion of NAD^+^.

## 4. Materials and Methods

### 4.1. Neurite Length Measurement

DRG neurons from adult mice were cultured as described [72,73]. In brief, the plate wells were first coated with 250 µL of poly-L-lysine (PLL; 100 µg/mL in water), then laminin (200 µL of 2.5 µg/mL in PBS). DRGs were collected from 3–4-month-old adult WT or nSIRT1OE Tg mice, digested with papain and collagenase to dissociate the DRG neurons and cultured in SHTE medium (Selenium 5.2 µg/mL, Hydrocortisone 7.6 µg/mL, Transferrin 10 µg/m, Estradiol 5.4 µg/mL) as described [73]. A specific amount of glucose was added to the SHTE medium (control = 5.5 mM, high glucose = 30 mM glucose).

Immortalized rat DRG neurons, 50B11, were plated on a 0.01% poly-l-lysine-coated 24-well tissue culture plate and treated with forskolin (final concentration was 75 µM) in the absence or in the presence of NR (50 μg/mL) or NMN (100 μg/mL) for 72 h. Subsequently, differentiated 50B11 cells wells were gently overlaid with 2% glutaraldehyde in PBS for 20 min to fix the cells to the tissue culture plate. The glutaraldehyde solution was changed to 1% Coomassie Brilliant Blue G-250 (CBB) solution (1% CBB in 50% methanol/PBS), followed by staining for 2 h at room temperature. The cells were de-stained with 50% methanol/PBS and water, respectively.

Bright-field cell images were obtained using color phase-based images from a BXZ-700 microscopy (KEYENCE, Osaka, Japan) at 20-fold magnification. Neurite outgrowth was defined as the longest neurite extension in μm per neuron, and the average value was calculated for each culture (Figure 1)

### 4.2. Diabetes Induction with STZ

All animal protocols followed the National Institutes of Health (NIH) Guide for the Care and Use of Laboratory Animals and were approved by the Institutional Animal Care and Use Committee. Three-month-old, male, C57BL6 WT and Streptozotocin- (STZ-) induced diabetic mice were purchased from Jackson Labs. Baseline neuropathy measurements were taken in both groups (Time point 0), and then all mice were fed a control diet (Harlan Teklad #2018, West Lafayette, IN, USA) for 2 months. Diabetic mice having a fasting blood glucose level of 300 mg/dL (11 mmol/L) were used. At the end of 2 months, neuropathy measurements were made in all mice. Once diabetic neuropathy developed, the diabetic mice were split into three groups: diabetic mice that received NR in their diet (300 mg/kg) for an additional 2 months or diabetic mice that were injected with NMN (100 mg/kg) intraperitoneal (ip) on alternate days for an additional 2 months or diabetic mice that continued on their normal diet for an additional 2 months. The non-diabetic mice continued on the control diet for a total period of 4 months. At 4 months, NCSs, Von Frey, MA, and IENFD were performed in all mice.

### 4.3. CD and HFD

Three-month-old, wild-type, C57BL6 mice were fed with a CD or an HFD. The CD (Harlan Teklad #2018) contained 6.2% fat (18% calories from fat), 18.6% protein (24% calories from protein), and 44.2% carbohydrate (58% calories from carbohydrate). The HFD (Bio-Serv #F3282, Flemington, NJ, USA) contained 36% fat (60% calories from fat), 20.5% protein (15% calories from protein), and 37.5% carbohydrate (26% calories from carbohydrate). Baseline NCSs were conducted at the beginning of the study. The SMNCV, TML, TSNCV, and MA were measured in the CD and the HFD mice at 1 and 2 months. After confirming that consumption of an HFD for 2 months induced the development of peripheral neuropathy as observed by the changes in the NCSs and MA, the HFD mice were split into three groups: HFD mice that received NR in their diet (300 mg/kg) for an additional 2 months, HFD mice that were injected with NMN (100 mg/kg) intraperitoneal (ip) on alternate days for an additional 2 months, and diabetic mice that continued on their HFD diet for an additional 2 months. Meanwhile, the CD mice continued the CD for a total period of 4 months. At 4 months, neuropathy measures were performed in all mice.

### 4.4. Quantification of NAD^+^ and SIRT1 Activity

The NAD^+^ level was quantified following the protocol of Liang et al. [74] and as previously described [1]. Briefly, NAD^+^ analysis in tissue samples was carried out with a Shimadzu Nexera coupled to a QTRAP^®^ 5500 mass spectrometer (AB Sciex, Framingham, MA, USA) equipped with a turbo–electrospray interface in positive ionization mode. The aqueous mobile phase was water with 0.1% formic acid, and the organic mobile phase was acetonitrile with 0.1% formic acid. The gradient was 0% formic acid for the first 0.1 min, and then increased to 30% formic acid in 0.9 min, decreased to 0% formic acid within 0.1 min, and maintained at 0% formic acid for another 0.4 min. The flow rate was 0.8 mL/min and the cycle time (injection to injection including instrument delays) was approximately 1.8 min. A volume of 1–3 μL of the final extract was injected onto the analytical dC18 column (100 × 2.1 mm, 3 μm, Waters, Milford, MA, USA).

SIRT1 activity was measured in the DRG tissue extracts following the SIRT1 Activity Assay Kit (Fluorometric; ab 156065, Abcam, Waltham, MA, USA). Using the kit, the fluorophore and quencher were coupled to the amino terminal and carboxyl terminal of the substrate peptide, respectively, allowing fluorescence emission. If SIRT1 performs deacetylation, the substrate peptide is cut through the action of the protease added simultaneously, and the quencher will separate from the fluorophore, allowing emission of fluorescence. Deacetylase enzyme activity was measured by measuring the fluorescence intensity. Protein extraction and Western blot analysis was conducted as described [3].

### 4.5. Neuropathy Measurements

Peripheral neuropathy was tested following the guidelines of the European diabetic neuropathy study group of the EASD (Neurodiab) [75]. Mechanical allodynia was assessed using Somedic von Frey monofilaments [3,72]. Nerve conduction studies were performed as described [3,76]. The IENFD was measured using PGP9.5 antibody staining in a blinded fashion, as previously described [3,72]. The IENFD was calculated (as fibers/mm) by the number of complete baseline crossings of nerve fibers at the dermo–epidermal junction divided by the measured length of the epidermal surface using standardized validated methods [77,78].

### 4.6. Statistical Analysis

A comparison of dependent variables was performed on transformed data using factorial ANOVA with a post hoc Tukey test to determine the significance among the groups. Individual comparisons were made using Student’s *t*-test, assuming unequal variances as previously described [79]. The associations between mitochondrial function and measures of neuropathy (NCVs and mechanical allodynia) were evaluated using Spearman correlation statistics.

## Figures and Tables

**Figure 1 ijms-25-01102-f001:**
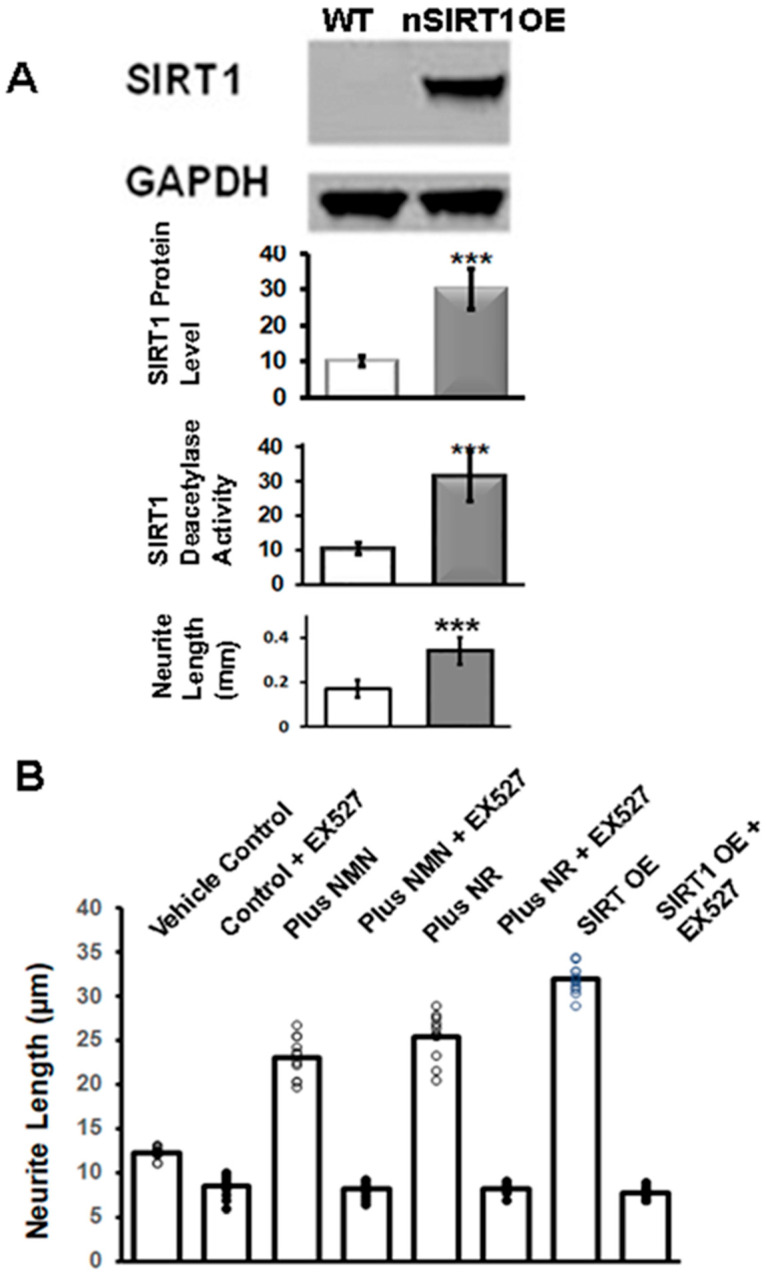
Addition of NMN or NR or SIRT1OE promoted neurite growth in cultured DRG neurons in a SIRT1-dependent manner. (**A**) Primary DRGs were cultured from 3-month-old nSIRT1OE mice and wild-type (WT) mice for 4 days and the length of neurite outgrowth was measured. Neurite outgrowth was defined as the longest neurite extension in μm per neuron, and the average value was calculated for each culture. Protein extracts were prepared, and the levels of SIRT1 protein were measured with a Western blot analysis and the SIRT1 activity was measured via fluorometric assay [3]. (**B**) Cultured immortalized rat DRGs (50B11) were differentiated with forskolin (75 µM) in the absence or in the presence of added NR (150 μg/mL) or NMN (100 μg/mL). 50B11 cells were transfected with CMV-SIRT1 over expression (OE) vector. Duplicate cultures were also treated with SIRT1 inhibitor EX527 (5 μM). Neurite length was measured 72 h after the addition of NR/NMN/EX527 or after transfection with SIRT1OE vector. The addition of NMN or NR increased neurite length compared to vehicle control (*p* < 0.01), and transfection with SIRT1OE vector also increased neurite length (*p* < 0.001). Addition of EX527 inhibited forskolin-induced neurite growth. (*p* < 0.001). *** = *p* < 0.001 in subfigure A; Circle represent individual experimental value in subfigure B.

**Figure 2 ijms-25-01102-f002:**
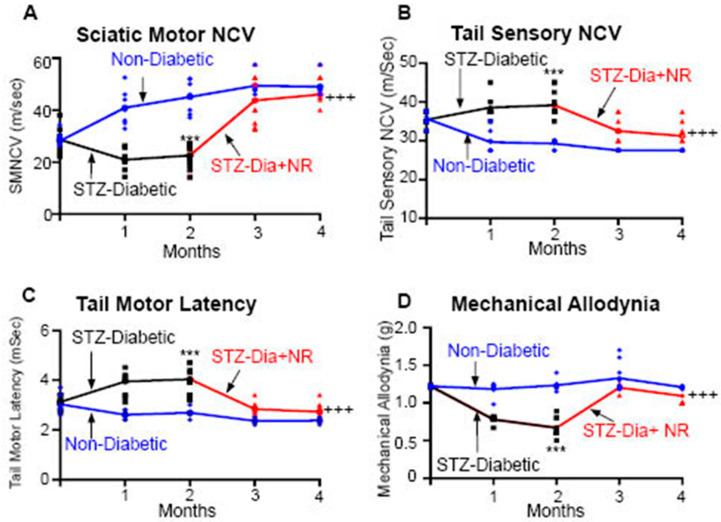
NR reverses STZ-induced neuropathy in C57BL6 mice (n = 6/group). Three-month-old C57BL6 WT and Streptozotocin- (STZ-) induced diabetic mice were purchased from Jackson Labs. Baseline nerve conduction studies (NCSs) were completed at the beginning of the study. Sciatic motor nerve conduction velocity (SMNCV; (**A**)), tail sensory nerve conduction velocity (TSNCV; (**B**)), tail motor latency (TML; (**C**)), and mechanical allodynia (MA; (**D**)) were measured in non-diabetic (control) and in STZ mice at 1 and 2 months. After confirming that the STZ mice had developed neuropathy as observed by the changes in the NCSs and MA, the STZ mice were fed with NR in the diet at a dose of 300 mg/kg for an additional 2 months. NCSs were performed at 3 and 4 months, namely after 1 and 2 months of the administration of NR. The results are shown as follows: sciatic SMNCV (**A**), TSNCV (**B**), TML (**C**), and MA (**D**) via the Von Frey filament paw withdrawal threshold method. Statistical comparisons were made among the three groups with the ANOVA and post hoc Tukey test. *** *p* < 0.001; STZ at 2 months compared to 0-month-old STZ and non-diabetic mice in all parameters. ^+++^
*p* < 0.001, STZ + NR at 4 months compared to STZ at 2 months in all parameters. Dietary administration of NR reversed all the deficits of STZ-induced neuropathy. Administration of NR to non-diabetic mice had no significant effect [1].

**Figure 3 ijms-25-01102-f003:**
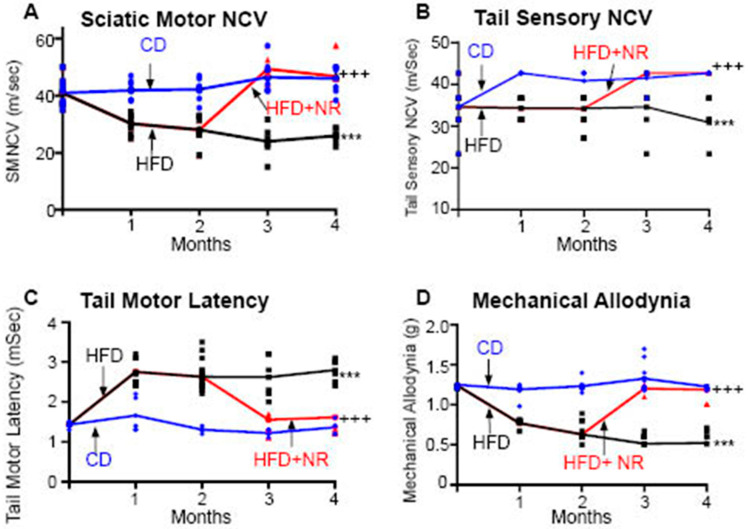
NR reverses HFD-induced neuropathy in C57BL6 mice (n = 6/group). Three-month-old, male, C57BL6 WT mice were fed with either a control diet (CD) or a high fat diet (HFD) for 2 months. Baseline NCSs were completed at the beginning of the study. SMNCV, TML, TSNCV, and MA were measured in the mice fed a CD and the mice fed an HFD at 1 and 2 months. After confirming that consumption of the HFD for 2 months induced development of peripheral neuropathy as observed by the changes in the NCSs and MA, NR was added to the HFD mice at a dose of 300 mg/kg for an additional 2 months. Nerve conduction studies were performed after 1 and 2 months of the administration of the NR. SMNCV (**A**), TML (**B**), TSNCV (**C**), and mechanical allodynia via the Von Frey filament paw withdrawal threshold method (**D**). Statistical comparisons were made among the three groups with the ANOVA and post hoc Tukey test. *** *p* < 0.001; HFD mice at 2 months compared to 0-month-old HFD and CD mice in all parameters. ^+++^
*p* < 0.001, STZ + NR mice at 4 months compared to STZ at 2 months in all parameters. The dietary administration of NR reversed all the deficits of STZ-induced neuropathy. The administration of NR to non-diabetic mice had no significant effect [1].

**Figure 4 ijms-25-01102-f004:**
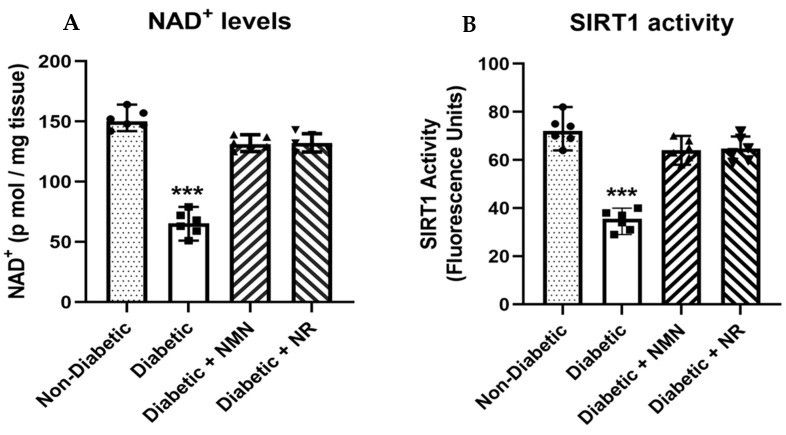
The NAD^+^ and SIRT1 activity levels were decreased in the DRGs from the STZ diabetic animals. NMN treatment repairs the NAD^+^ deficit, increasing SIRT1 activity. (**A**) Perchloric acid extraction of the frozen DRG tissue with added internal standard of NAD^+^ was completed as described [1]. NAD^+^ analysis in tissue samples was carried out with QTRAP^®^ 5500 mass spectrometer (AB Sciex, Framingham, MA, USA) equipped with a turbo–electrospray interface in positive ionization mode. The results showed a significant decrease in NAD^+^ levels in diabetic DRG neuronal tissue, and the treatment of diabetic mice with NMN or NR increased the NAD^+^ levels to tissue levels in non-diabetic DRGs. (**B**) SIRT1 activity was measured in the DRG tissue extracts following the SIRT1 Activity Assay Kit (Fluorometric; ab 156065). The results show a significant decrease in SIRT1 activity in diabetic DRG neuronal tissue, and the treatment of diabetic mice with NMN or NR increased the SIRT1 activity to tissue levels in non-diabetic DRGs. *** = *p* < 0.001; Triangles, circles and squares represent individual sample values.

**Figure 5 ijms-25-01102-f005:**
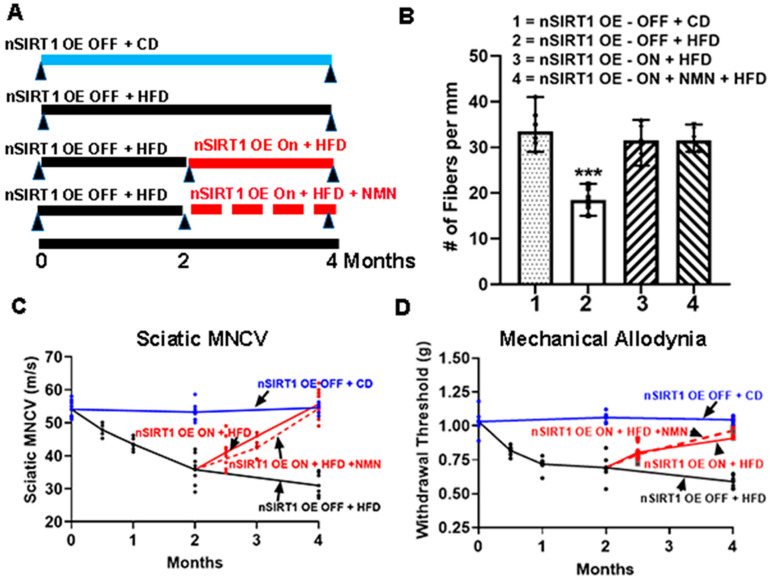
NMN administration does not enhance reversal of HFD neuropathy in nSIRT1OE mice (n = 6/group). The nSIRT1OE mice were generated as previously described [3]. The expression of nSIRT1 was shut off by feeding the bigenic mouse with doxycycline (DOX, 200 mg/kg diet). Three-month-old, DOX-fed, male, nSIRT1OE-OFF mice were fed with either a control diet (CD; Group #1) or a high fat diet (HFD; Group #2) for 4 months. Baseline NCSs were performed at the beginning of the study. The SMNCV, TML, TSNCV, and MA were measured in the CD mice and the HFD mice at 2 and 4 months. After confirming the induction of peripheral neuropathy in the HFD, nSIRTOE-OFF mice at 2 months, the nSIRT1OE was tuned on by removing the DOX from the diet. The nSIRT1OE mice were divided into two groups: Group #3 was fed with a DOX-free HFD for an additional 2 months. Group #4 was fed with a DOX-free HFD, and NMN was administered on alternate days at a dose of 100 mg/kg ip for an additional 2 months. NCSs were performed after 2 weeks and after 2 months of the administration of NMN. The model for the experiment (**A**), IENFD (**B**), SMNCV (**C**), mechanical allodynia through the Von Frey filament paw withdrawal threshold method (**D**). There was no significant difference with NMN administration in the nSIRT1OE mice compared to nSIRT1OE alone. *** = *p* < 0.001; the # represents the groups that were compared.

**Table 1 ijms-25-01102-t001:**
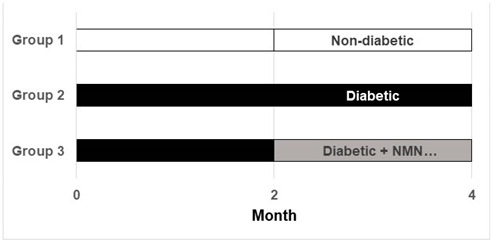
NMN reverses STZ-induced neuropathy in C57BL6 mice (n = 6/group). Three-month-old C57BL6 WT and Streptozotocin- (STZ-) induced diabetic mice were purchased from Jackson Labs. Baseline NCSs were completed at the beginning of the study. SMNCV, TML, TSNCV, and MA were measured in the non-diabetic (control) and the STZ mice at 1 and 2 months. After confirming that the STZ mice had developed neuropathy as observed by the changes in the NCSs and MA, the STZ mice were injected with NMN (100 mg/kg) intraperitoneal (ip) on alternate days for an additional 2 months. The schematic approach is shown in Table 1. The results of the NCSs are shown in Table 1. Statistical comparisons were made among the three groups. The administration of NMN reversed all the deficits of the STZ-induced neuropathy. The administration of NMN to non-diabetic mice had no significant effect [1].

Groups	Non-Diabetic	Diabetic	Diabetic + NMN	Significance *p* Values		
Group #	1	2	3	1 vs. 2	2 vs. 3	1 vs. 3
SMNCV (m/s)	48 ± 7	29 ± 6	44 ± 6.5	<0.001	<0.001	NS
TML (m/s)	1.3 ± 0.2	2.2 ± 0.3	1.4 ± 0.3	<0.001	<0.001	NS
TSNCV (m/s)	44 ± 3	25 ± 5	42 ± 6	<0.001	<0.001	NS
Von Frey MA	1.2 ± 0.2	0.6 ± 0.1	1.2 ± 0.3	<0.001	<0.001	NS
IENFD #/mm	22 ± 3	12 ± 2	21 ± 3	<0.001	<0.001	NS

**Table 2 ijms-25-01102-t002:**
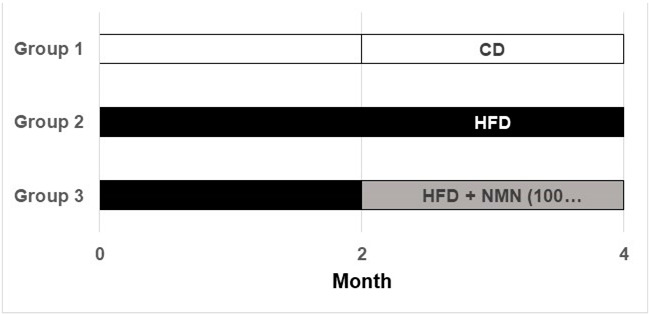
NMN reverses HFD-induced neuropathy in C57BL6 mice (n = 6/group). Three-month-old, male, C57BL6 WT mice were fed with either a control diet (CD) or a high fat diet (HFD) for 2 months. Baseline NCSs were completed at the beginning of the study. The SMNCV, TML, TSNCV, and MA were measured in the non-diabetic (control) and in the HFD mice at 2 months.

Groups	CD-Fed	HFD-Fed	HFD-Fed + NMN	Significance *p* Values		
Group #	1	2	3	1 vs. 2	2 vs. 3	1 vs. 3
SMNCV (m/s)	49 ± 6	31 ± 6	45 ± 6.5	<0.001	<0.001	NS
TMl (m/s)	13 ± 0.3	2.4 ± 0.3	1.4 ± 0.3	<0.001	<0.001	NS
TSNCV (m/s)	41 ± 3	31 ± 5	42 ± 6	<0.001	<0.001	NS
Von Frey MA	1.2 ± 0.2	0.6 ± 0.1	1.2 ± 0.3	<0.001	<0.001	NS
IENFD #/mm	27 ± 6	12 ± 4	24 ± 4	<0.001	<0.001	NS

**Table 3 ijms-25-01102-t003:** Comparison of treatment with and without NMN in the NCSs of the nSIRT1OE mice at the conclusion of the study. Statistical comparisons were made among the four groups. Group # is Group Number. Group#1: nSIRT1OE-OFF + CD; Group #2: nSIRT1OE-OFF + HFD; Group #3: nSIRT1OE-ON + HFD; and Group #4: nSIRT1OE-ON + HFD + NMN) via the ANOVA and post hoc Tukey test. There was a statistical difference between the CD and HFD mice (Group #1 vs. Group #2). There was no statistical difference between the nSIRT1OE alone or with NMN (Group #3 vs. Group #4).

Parameters	nSIRT1OE-OFF		nSIRT1OE-ON		Significance			
	CD (n = 6)	HFD (n = 6)	HFD (n = 6)	HFD + NMN (n = 6)	1 vs. 2	2 vs. 4	1 vs. 3	3 vs. 4
Group #	1	2	3	4				
SMNCV (m/s)	45.2 ± 7.7	36.6 ± 8.2	46.3 ± 4.7	47.5 ± 7.1	<0.001	<0.001	NS	NS
TML (m Sec)	1.5 ± 0.12	2.2 ± 0.19	1.4 ± 0.14	1.4 ± 0.12	<0.001	<0.001	NS	NS
TSNCV (m/s)	33.9 ± 2.5	27.3 ± 3.6	35.5 ± 4.9	35.6 ± 1.4	<0.001	<0.001	NS	NS
Von Frey (g)	11 ± 0.2	0.4 ± 0.2	1.2 ± 0.12	1.0 ± 0.3	<0.001	<0.001	NS	NS
Hargreaves (sec)	8.4 ± 1	12 ± 1.9	8.6 ± 1	7.8 ± 1.4	<0.001	<0.001	NS	NS

## Data Availability

Data are available on request.

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
