# Peer review of "NAD+ Precursors Reverse Experimental Diabetic Neuropathy in Mice"

_ijms, 2024, doi:10.3390/ijms25021102_

Round 1

Reviewer 1 Report

Comments and Suggestions for Authors

I have thoroughly reviewed your manuscript and find it to be an intriguing study with valuable insights. I would gladly endorse its publication with a few considerations for revision.

Firstly, I recommend that in line 38, the term "NAD+" be spelled out when it is initially introduced in the manuscript. This ensures clarity for readers who may not be familiar with the abbreviation.

Secondly, it would enhance the manuscript if the aim of the study is explicitly stated at the end of the introduction. Clearly articulating the study's objective will provide readers with a comprehensive understanding of the research from the outset.

Thirdly, I suggest that the authors include a statement outlining the hypothesis of the study. This addition will contribute to the overall coherence of the manuscript and aid readers in grasping the scientific context and objectives.

Fourthly, in Figure 1, there is a reference to "GAPDH." It would be beneficial to provide an explanation or spell out the meaning of this abbreviation for readers who may not be familiar with it. This clarification will improve accessibility and understanding of the graphical representation.

Fifthly, concerning the subtitle in section 2.2, I recommend spelling out "STZ" to avoid potential confusion for readers who may not be well-versed in the field. This small adjustment will contribute to the manuscript's overall accessibility.

Lastly, in Table 3, I suggest reconsidering the addition of a "#" behind the group. It would be helpful to provide clarity on the rationale for this symbol and whether it is essential for readers' comprehension of the data.

Reviewer 2 Report

Comments and Suggestions for Authors

The article is interesting and well-written. I only have minor comments/questions (listed below):

1. In the Introduction section, I think an additional paragraph that explains the author’s choice of using NMN and NR for their study would be useful for the reader, especially since a lot of information about SIRT1 is provided (and little information about NMN and NR).

2. There are several comments and literature references in the Results section. I suggested clearly delineating Results from Discussions - presenting the results in an objective manner, with as little information regarding the methods used as possible, and then discussing the results in a distinct section.  
